# Long-Term Ambient Air Pollutant Exposure and Risk of Recurrent Headache in Children: A 12-Year Cohort Study

**DOI:** 10.3390/ijerph17239140

**Published:** 2020-12-07

**Authors:** Syuan-Yu Hong, Lei Wan, Hui-Ju Lin, Cheng-Li Lin, Chang-Ching Wei

**Affiliations:** 1Division of Pediatric Neurology, Department of Pediatrics, Children’s Hospital, China Medical University Hospital, Taichung 40447, Taiwan; D28320@mail.cmuh.org.tw; 2School of Chinese Medicine, China Medical University, Taichung 40447, Taiwan; lei.joseph@gmail.com (L.W.); d2396@mail.cmuh.org.tw (H.-J.L.); 3Management Office for Health Data, China Medical University Hospital, Taichung 40447, Taiwan; orangechengli@gmail.com; 4Institute of Biostatistics, China Medical University, Taichung 40447, Taiwan; 5Children’s Hospital, China Medical University Hospital, Taichung 40447, Taiwan; 6School of Medicine, China Medical University, Taichung 40447, Taiwan

**Keywords:** air pollution, headache, migraine

## Abstract

Although studies have suggested environmental factors to be triggers of headache, the contribution of long-term air pollution exposure to recurrent headaches is poorly understood. Hence, we executed this nationwide cohort study to investigate associations between levels of ambient air pollutants and risks of recurrent headaches in children in Taiwan from 2000 to 2012. We used data from the Taiwan National Health Insurance Research Database and linked them to the Taiwan Air Quality Monitoring Database. Overall, 218,008 children aged < 18 were identified from 1 January 2000, and then followed until they were diagnosed by a physician for ≥3 times with recurrent headaches or until 31 December 2012. We categorized the annual average concentration of each air pollutant (fine particulate matter, total hydrocarbon, methane, sulfur dioxide, and nitrogen dioxide) into quartiles (Q1–Q4). We measured the incidence rate, hazard ratios (HRs), and the corresponding 95% confidence intervals for recurrent headaches. stratified by the quartiles. A total of 28,037 children (12.9%) were identified with recurrent headaches. The incidence rate and adjusted HR for recurrent headaches increased with higher-level exposure of air pollutants, except sulfur dioxide. We herein demonstrate that long-term ambient air pollutant exposure might be a risk factor for childhood recurrent headaches.

## 1. Introduction

Children with recurrent headaches result in great impact on their life but also on their families [1,2]. Migraines, characterized by recurrent headaches, are among the most common childhood headaches [1,2]. Furthermore, childhood migraine may persist into adulthood [3]. The influence of environmental factors on the attacks of migraine/recurrent headaches leads to extensive debate over the past decades [4]. Determining the triggering factors of headaches is crucial to prevent the illness. Chemical exposure and specific environmental irritants are well-known triggers of headaches [4]. Air pollution is the most widespread and inevitable form of pollution, and it may contribute to serious short-term and long-term health effects [5], involving a large variety of ailments and conditions that include chronic obstructive pulmonary disease, neurobehavioral disorders, lung cancer, birth defects, leukemia, premature death, asthma, immune system defects, and cardiovascular diseases [6,7,8]. Patients with migraine/headaches often complain that poor air quality aggravates or triggers their headaches [9]. Indeed, environmental factors, such as air pollutants and weather, may produce neurogenic inflammation and trigger migraine/recurrent headaches onset [10,11]. 

To date, several studies have suggested an association between some outdoor air pollutants and frequency, severity, or medical consultation rates for headaches [12,13,14]; nevertheless, among the mentioned studies, the majority have accentuated adult patients and short-term influences induced by air pollution on migraine/recurrent headaches [12,13,14]. Most of our current knowledge about migraine/recurrent headaches in children is based on extrapolations from studies conducted on adults. Children and adolescents are notably different from adults in terms of their rapid growth, significant neurological development, and psychological changes. Thus, the long-term influences engendered by air pollution on recurrent headaches should be examined for the pediatric age group specifically. 

This nationwide cohort study entailed employing the Taiwan National Health Insurance Research Database (NHIRD) and the Taiwan Air Quality Monitoring Database (TAQMD) for probing the long-term influences induced by outdoor air pollution on the incidence rates and risk of childhood recurrent headaches. The findings may promote informed practical clinical perspectives and advance public awareness of the negative influences that are induced by air pollution on children’s health.

## 2. Materials and Methods 

### 2.1. Data Source

The Taiwan National Health Insurance (NHI) program is known to offer coverage to 99% of Taiwan’s 23 million residents and to contains information regarding contracts with over 90% of the region’s national health care facilities (https://nhird.nhri.org.tw/en/index.html) [15,16]. The corresponding electronic database of this program, namely the NHIRD, comprises the claims data of insured population. Published studies have validated the high reliability of NHIRD diagnostic data [15,16]. The NHIRD includes detailed information, such as outpatient visits, hospital admissions, prescriptions, procedures, and disease diagnoses executed on the basis of the International Classification of Diseases, Ninth Revision, Clinical Modification (ICD-9-CM) [17]. An exclusive personal identification number (PIN) is assigned to every individual in Taiwan. In the NHIRD, for patient privacy protection, data concerning patient identities are scrambled cryptographically [15,17]. The entirety of NHI data sets can also be cross-referenced with each individual’s PIN. This study utilized an NHIRD-derived data file, namely the Children file, that comprises information from half (chosen at random) of all insured children in Taiwan [18]. The data set was determined to afford an adequate sample for pursuing the study objectives. This study was ratified by Research Ethics Committee of China Medical University and Hospital (CMUH104-REC2-115), and it complied with the principles outlined in the Declaration of Helsinki. This study received approval from the institutional review board of China Medical University Hospital (CRREC-103-048).

### 2.2. Study Population, Outcome of Interest, Endpoints, and Confounding Factors

This was a retrospective cohort study. From the Children file, we formed a child cohort by selecting individuals aged <18 (0–17) years on 1 January 2000 (baseline). The study period was from 1 January 2000 to 31 December 2012. The follow-up period of each participant began from baseline until recurrent headaches onset, end of the follow-up, termination of insurance, or death. Individuals who were missing information such as their address, sex, and air pollution data and individuals that had ever been diagnosed with migraine before the baseline were excluded (Figure 1). National Health Insurance (NHI) data is a useful tool for massive epidemiological investigation, but it has some systemic problems regarding accuracy when it targets to specific disease (especially disease diagnosed by clinical diagnostic criteria, e.g., migraine). The clinical diagnostic diseases obtained from NHI data were coded with ICD-9-CM by physicians individually, and most of them were not specialists. Migraine, characterized by recurrent headaches, might be diagnosed by physicians who were not specialists in Neurology and intend to describe “recurrent headaches” in their clinical setting. In view of this, we defined migraine/headaches in the present study as ≥3 times diagnoses of ICD-9-CM code 346 and/or 784.0 in any diagnosis field during any inpatient or ambulatory claim process during study period. Despite less than ideal, we believe this is the best method to identify our target patients. Figure 1 shows the flowchart of study design and study population selection.

By the end of the study period, some participants would be entering adulthood. The final study population contained 218,008 participants. Our mean (standard deviation) follow-up years for patients with recurrent headaches was 10.7 (2.6). Urbanization level of residence, age, number of consultations/visits with a physician per year, parental monthly income, sex, and allergic diseases constituted the confounding factors. On the basis of the method realized by Liu et al. [19], we classified the study patients’ residential areas, encompassing 365 townships of Taiwan, into seven urbanization levels, with Levels 1 and 7 representing the “most urbanized” and “least urbanized,” respectively. We stratified the townships for defining urbanization levels by using several variables, including the following: population ratio of people with an educational level of college or higher, population density (people/km^2^), population ratio of agricultural workers, number of physicians per 100,000 people, and population ratio of elderly people aged older than 65 years [19]. Because Levels 4–7 were determined to have low sample size, we combined these four levels into a single group (Level 4). Thus, we stratified the factor urbanization level into four levels, with Levels 1 and 4 representing the highest density and lowest density, respectively. We also classified monthly income into the following three groups: >NT$20,000, NT$15,000–19,999, and <NT$15,000.

### 2.3. Exposure Measurement

Ambient air monitoring of monthly average data for SO_2_, NO_2_, THC, CH_4_, and PM_2.5_ were collected from 74 ambient air quality-monitoring stations distributed all over Taiwan during 1998–2012. Concentrations of each pollutant are measured hourly—CO by nondispersive infrared absorption, NO_2_ by chemiluminescence, SO_2_ by ultraviolet fluorescence, THC and CH_4_ by flame ionization detector, and PM_2.5_ by beta-gauge—and are reported hourly.

We identified the map coordinates of the monitoring stations and air pollution sources. The ultraviolet fluorescence in these recording stations was automatically monitoring and recording readings of PM_2.5_, THC, CH_4_, SO_2_, and NO_2_. The daily air pollution data were averaged based on these recording stations. Yearly average concentrations of pollutants were calculated from the baseline to the date of recurrent headaches occurrence, the withdrawal of patients, or the end of the study period, and the data were categorized into quartiles. The participants were assigned to residential districts based on the clinic where they most frequently sought treatment for acute upper respiratory infection (ICD-9-CM code 460). We divided the annual average air pollutant concentrations into quartiles: Q1, Q2, Q3, and Q4. We categorized annual average PM_2.5_ into Q1 (<11,120 μg/m^3^), Q2 (11,120–12,652 μg/m^3^), Q3 (12,652–15,056 μg/m^3^), and Q4 (>15,056 μg/m^3^); THC into Q1 (<835 ppm), Q2 (835–877 ppm), Q3 (877–949 ppm), and Q4 (>949 ppm); CH_4_ into Q1 (<735 ppm), Q2 (735–754 ppm), Q3 (754–770 ppm), and Q4 (>770 ppm); SO_2_ into Q1 (<1346 ppb), Q2 (1346–1914 ppb), Q3 (1914–2338 ppb), and Q4 (>2338 ppb); NO_2_ into Q1 (<7896 ppb), Q2 (7896–8894 ppb), Q3 (8894–10,214 ppb), and Q4 (>10,214 ppb).

The air pollutant measurements from Taiwan EPA monitoring stations were integrated into monthly point data and interpolated to pollutant surfaces using inverse distance weighting (IDW). For the IDW approach, we used inverse squared distance (1/squared distance) weighted average of the three nearest monitors to compute monthly mean concentration. IDW predicts values of unknown points based on the similarity of two objects by its distance. When the unknown point to be estimated is closer to the known measuring point, the weighted value of the unknown point will be higher. We used the air pollution exposure in the two years before and the current year of diagnosis of headaches to predict the monthly air pollution and used IDW method to estimate the air pollution concentrations between the measured values of the air monitoring stations around the household registered by each patient according to the distance. Then, we explored the association between air pollutant and headaches. (All data were managed by a geographic information system (ArcGIS version 10.3; ESRI, Redlands, CA, USA)).

### 2.4. Statistical Analysis

The sociodemographic factors in the current study included residential area urbanization level, sex, parental monthly income, age, and daily average exposure to air pollutants. To test the differences in daily average concentration distributions for each air pollutant by quartile and urbanization, we executed χ^2^ testing. Moreover, we calculated the incidence density rate of recurrent headaches (per 1000 person–years) according to each quartile of daily average concentrations for the five air pollutants. By employing Cox proportional hazard regression, we also derived estimates of the hazard ratios (HRs) as well as 95% confidence intervals (CIs) corresponding to recurrent headaches at the Q2–Q4 levels for air pollutant concentrations relative to the lowest level (Q1). To address the concern of constant proportionality, we examined the proportional hazard model assumption using a test of scaled Schoenfeld residuals. In the model evaluating the recurrent headaches risk throughout overall follow-up period, results of the test revealed a significant relationship between Schoenfeld residuals for PM_2.5_, THC, CH_4_, SO_2_, and NO_2_ and follow-up time (*p*-value < 0.001, respectively), suggesting the proportionality assumption was violated. To deal with non-proportional hazards, we were used extended Cox models with time-dependent terms shows results. 

We adjusted the applied multivariable model for allergic diseases, sex, number of consultations/visits with a physician per year, urbanization level, age, and monthly income. We also added the exposures as a continuous variable to estimate the risk of recurrent headaches as sensitivity testing. Further, we calculate the month average air concentration to estimate the month exposed air concentration for each patient by Inverse Distance Weighting Method (IDW methods). The IDW method is one of the most commonly used spatial interpolation methods in Geosciences, which calculates the prediction values of unknown points by weighting the average of the values of known points (these data were analyzed with ArcGIS version 10.3). We accessed the air pollution in the 2 years before and the current year of the diagnosis of recurrent headaches and used IDW method to estimate the air pollution concentrations between the measured values of the air monitoring stations around the household registered by each patient according to the distance as sensitivity testing. The Statistical Package for the ArcGIS version 10.3 as well as SAS 9.3 (SAS Institute Inc, Cary, NC, USA) constituted the platforms for all the executed analyses in this study. Additionally, for all executed statistical analyses, we deemed 2-tailed *p* values of <0.05 to indicate statistically significant tests.

## 3. Results

During the study period, a total of 28,037 children (12.9%) were diagnosed with recurrent headaches in the cohort of 218,008 children. Table 1 presents the participants’ sociodemographic factors. The mean age of participants was 6.01 years (standard deviation, 2.98). The mean follow-up period was 10.7 years (standard deviation, 2.6). The proportion of boys was slightly higher than that of girls (52% vs. 48%), that is, similar to national demographic data reported by the Taiwan Ministry of Internal Affairs (the ratio of male to female under 15 is about 1.09:1). In addition, most participants lived in densely populated areas (65.9%).

The incidence rates for recurrent headaches increased with higher levels of PM_2.5_, THC, CH_4_, and NO_2_ exposure (Table 2). The increasing incidence rates from Q1 to Q4 in PM_2.5_, THC, CH_4_, and NO_2_ were from 8.62 to 15.0, from 8.06 to 16.3, and from 8.93 to 24.1, and from 11.8 to 12.9 per 1000 person–years, respectively. The adjusted HR for recurrent headaches increased with higher levels of exposure in PM_2.5_, THC, CH_4_, and NO_2_. 

Further data analysis compared risk differences by sex and age (Table 3). The trends of dose–response relationships between pollutants and hazards of recurrent headaches were consistent between boys and girls and between young children and older ones. The risk might be higher for boys and children older than 6 years, while exposed to higher level air pollutants.

The Kaplan–Meier plots (Figure 2A–E) with pollutant concentration stratified by quartiles showed that patients exposed to higher pollution concentrations had higher accumulative incidence of recurrent headaches than those exposed to lower pollution concentrations of fine PM_2.5_, THC, CH_4_, SO_2_, and NO_2_. 

The distribution of air pollution exposures during the follow up time was shown in Figure 3A–E. 

In the interaction analyses for the risk of recurrent headaches between ambient temperature and concentration of ambient air pollutants, we found that the highest incidence rate and the highest adjusted HR were in those exposed ≥median temperature and ≥median-level of pollutants (Table 4). 

We used annual average concentrations of air pollutants as continuous variables and used extended Cox models with time-dependent terms to consider non-linear relations and interaction between air pollutants and time in Table 5. Table 5 showed that the adjusted HR of recurrent headaches development was increased with five air pollutants concentration (PM_2.5_, THC, CH_4_, SO_2_, and NO_2_). Further analysis of the extended Cox models with time-dependent terms shows results, indicating that the strength of the association increased over time (HR (*p* value), for PM_2.5_, 1.000072 (*p* < 0.001); for interaction term of PM_2.5_ and time, 1.000016 (*p* < 0.001); for SO_2_, 1.0003 (*p* < 0.001); for interaction term of SO_2_ and time, 1.0001 (*p* < 0.001)), and reduced over time (HR (*p* value),for THC, 1.02 (*p* < 0.001); for interaction term of THC and time, 0.99 (*p* < 0.001); for CH_4_, 1.045 (*p* < 0.001); for interaction term of CH_4_ and time, 0.99 (*p* < 0.001); for NO_2_, 1.0005 (*p* < 0.001); for interaction term of NO_2_ and time, 0.99 (*p* < 0.001)). 

Appendix A showed that Inverse Distance Weighting (IDW) to speculate monthly average concentration of PM_2.5_, THC, CH_4_, SO_2_, and NO_2_. Air pollutant concentrations were grouped into four levels based on quartile. After controlling for risk factors, patients exposed in higher air pollutant concentrations had a significant higher risk of recurrent headaches than patients exposed in lower air pollutant concentrations which including SO_2_ and NO_2_. For SO_2_, relative to Q1 concentrations, the Q2 (adjusted HR = 2.14, 95%CI = 2.06–2.23 and adjusted HR = 1.39, 95%CI = 1.33–1.45), Q3 (adjusted HR = 3.59, 95%CI = 3.45–3.73 and adjusted HR = 1.67, 95%CI = 1.60–1.75), and Q4 (adjusted HR = 3.46, 95%CI = 3.33–3.60 and adjusted HR = 3.29, 95%CI = 3.15–3.44) concentrations had a significant higher risk of recurrent headaches. 

## 4. Discussion

Air pollution has become one of most significant worldwide environmental health risk factors [5,6,7,8]. Currently, Southeast Asia is the most polluted area worldwide, with 2.6 and 3.3 million deaths ascribed to outdoor and indoor air pollution, respectively [20,21]. Taiwan is in eastern Asia, near the most polluted area in the world. Previous studies have reported an association between air pollution and increased frequency and severity of migraine [12,13,14]. The prevalence of migraine in Taiwanese adolescents has risen over the past decade [22]. Motivated by these clinical, public health, and environmental concerns, as well as the lack of data on the childhood headaches–air pollution interaction, we executed the first nationwide study on the association of long-term exposure to air pollution with the incidence and risk of childhood recurrent headaches. The findings quantify how air pollution affects children’s health, indicating higher ambient temperature and ambient air pollutant exposure levels to be associated with increased incidence and risk of childhood recurrent headaches. Our study thus demonstrates that ambient air pollutant exposures are indeed associated with recurrent headaches in Taiwanese children. According to the information released by the Environmental Protection Agency of Taiwan, the domestic air pollution emissions are mainly from two categories: first, mobile pollution sources (transportation), such as NO_2_, lead, THC, CO and CH_4_; second, fixed pollution sources (industrial processes, power generation, or waste disposal), including suspended particulates (PM_2.5_). As a representative, we chose THC and CH_4_ out of mobile pollution sources and fine PM_2.5_ out of the fixed pollution sources.

Lewis (2009) assessed epidemiological research executed over the past 25 years on migraine headache in adolescents and children; the researcher detected 64 cross-sectional studies that involved a total of 227,249 subjects among 32 different countries. The overall mean prevalence values of headache and migraine were estimated to be 54.4% (95% CI 43.1–65.8) and 9.1% (95% CI 7.1–11.1), respectively [23]. A United States study found that up to 5% of the pediatric population were determined to endure migraine [24]. Another study executed in Sweden recruited 9000 school children for probing the migraine prevalence; the mentioned study reported that approximately 4% of the recruited children had migraine, with the average onset age being six years [25]. Furthermore, the migraine prevalence estimates for the ages of seven and 15 years were 1.4% and 5.3%, respectively [25]. In the present study, we defined recurrent migraine as visiting health care facilities more than 3 times because of migraine to avoid temporary headaches secondary to common cold or acute viral illness. Based on these criteria, the overall prevalence of migraine in children in Taiwan was 12.9%. The risk of migraine and recurrent headaches increased while exposed to higher level air pollutants in children older than 6. Although migraine is less common in children than in adults, it can begin in childhood and increase in prevalence with age. 

Airborne pollutants have long been regarded as environmental factors that trigger migraine/headaches. However, most previously published studies have accentuated the short-term influences engendered by air pollution on migraine/headaches [10,12,13,14]. The current state of knowledge suggests that a primary neuronal dysfunction leading to an increased sensitivity to a broad range of stimuli accounts for human migraine disorders [11,26]. Both genetic and environmental factors are likely to be pivotal to migraine phenomena [27,28]; a child’s brain develops in response to genes, the environment, and their interactions [29]. Although beyond the scope of the present study, future study exploring the interactions of gene, air pollution exposure, and migraine headaches is crucial to improve understanding of the mechanisms of childhood migraine/headaches.

Previously published studies regarding the influences exerted by air pollution on migraine headaches have offered inconsistent results. Chen et al. executed a similar investigation in 2015 and determined that high PM_2.5_ levels raise the risk of migraine-related clinic visits in Taipei (the capital city of Taiwan) [10]. Research in a Canadian population analyzed patients who visited emergency rooms for migraine between 1992 and 2002; the results showed PM_2.5_ to be associated with 3.3% increases in visits for migraine (95% CI: 0.6–6.0) as well as 3.4% increases in visits for headache (95% CI: 0.3–6.6) (12). Dales et al., (2009) studied seven Chilean urban centers; they observed an association between acute increases in ambient air pollution and increases in the number of headache-related hospital admissions [13]. However, Mukamal et al., (2009), who executed their work in Boston, did not find a clear association of air pollutants with risk of emergency room visits for migraine [14]. Our results support the association of long-term exposure to THC, PM_2.5_, and CH_4_ and high AMB TEMP to increased incidence and risk of childhood migraine.

Furthermore, our results indicate that most individuals living in areas with higher urbanization levels had higher air pollutant exposure, most likely due to the high vehicle density and population density in urban areas. Hence, the consumption of large amounts of energy and resources results in the emission of large amounts of air pollutants into the atmosphere in urban settings. On the other hand, although there are many sources of PM_2.5_, it is mainly from traffic, fixed pollution sources (industrial processes, power generation or waste disposal) and dust raised by public works (constructions), based on the statistics of the Environmental Protection Agency of Taiwan. Domestic data also revealed numerous large factories and heavy industries (such as cement, petrochemical, oil refining, steel and shipbuilding, and other related factories). In addition, due to the high degree of construction in Taiwan city, most public works have been transferred to suburbs or low population density areas in recent years. Therefore, the concentration of PM_2.5_ in these areas is relatively high. To date, the mechanism by which ambient temperature influences headache remains largely unknown. From a physiological view, headaches are associated with hemodynamic variations, and cold weather might aggravate such variations [30]. In addition, regarding both particulate air pollution and ambient temperature, PM_2.5_ was previously reported to be associated with migraine-related visits in the cold season, especially in female patients [12]. Additionally, during cold-front passages, the concentration of PM_2.5_ is relatively high [31]. Thus, it is inferred that when the temperature is low, it may cause poor vertical diffusion conditions and thus may result in a relatively high PM_2.5_ concentration under ordinary discharge conditions [31].

Our results reveal children who lived in areas with relatively a high ambient temperature had a relatively high accumulated incidence of recurrent headaches, in accordance with previously published studies [32]. Chen et al. and Roberts et al. [10,33] have emphasized the daily particulate air pollution–daily mean temperature interaction and gave several plausible explanations for this finding. In Taiwan, people have a higher likelihood of opening their windows or going outdoors in the warm season than they do in the cool season, leading to increased exposure; accordingly, monitored air pollutant concentrations could be more closely correlated with personal exposure in the warm season than in the cool season. However, extremely cold or extremely hot weather can be a triggering factor for migraine headache [34,35,36]. In Taiwan, located in the subtropical zone, summer temperatures are often above 38 °C, and annual average temperatures are often above 25 °C; hence, the effect of high temperatures is significantly higher than that of low temperatures.

The WHO ambient air quality guidelines suggest an annual mean PM_2.5_ concentration limit of 10 μg/m^3^ and 25 μg/m^3^ for the 24-hourly mean [37]. In the United States (US), the EPA reduces particle pollution by tightening the annual National Ambient Air Quality Standard for PM_2.5_ from 15 to 12 μg/m^3^ in 2012 [37]. Populations in large parts of the world, especially in East and Southeast Asia and the Middle East, are exposed to levels of fine particulate pollution that far exceed the WHO guidelines. In our study, the limit of the international guideline for PM_2.5_ (25 μg/m^3^) is only within the first quartile of the current study. Hence, most of our population exposed to levels of PM_2.5_ that far exceed the WHO guidelines. Taiwan is in east Asia, near the most polluted area of the world. The prevalence of migraine in Taiwanese adolescents has risen over the past decade. Our findings show ambient air pollutant exposure levels to be associated with increased incidence and risk of childhood recurrent headaches.

On the other hand, this study did not reveal positive association between ambient concentration of SO_2_ and risks for recurrent headaches in children. Interestingly, Szyszkowicz et al. conducted a study in Canada and found that SO_2_ exposure had a positive correlation for migraine in female adults during warm seasons [38]. However, another two studies in Asia did not find significant association between SO_2_ exposure and migraine development, whereas the other air pollutants were found to trigger migraine particularly on relatively hot and cool days [39,40]. Based on the above, we speculate because of the nature of sulfur dioxide—a molecular compound that is heavier than air and whose chemical properties are highly related to temperature and climate. Therefore, people who live in high latitude areas with continental climate, year-round low temperature days, and poorly ventilated and enclosed environments are susceptible to the harmful influence of SO_2_ exposure. Taiwan is probably spared the damage because of its island type subtropical climate. However, further studies are needed to confirm this presumption.

Potential limitations that could serve as confounders to this study should be acknowledged. First, genetic and environmental factors (such as stress, family member’s smoking habits, physical activity, dietary habits, family history of migraine, and emotional factors), severity of migraine, and subtype of migraine were not captured in administrative claims databases. Second, children with high exposure to air pollution lived in areas with relatively high urbanization levels. Children living in relatively crowded environments might have relatively high stress associated with notably crowded housing. Moreover, children in highly polluted areas may have numerous diseases other than migraine/headaches; their consultations for respiratory tract infections and allergic diseases may be more frequent than the national average; the more frequently a child consults the doctor, the more opportunities that child has of being diagnosed with migraine/headaches. Thus, we adjusted for possible confounders such as numbers of consultations/visits with a physician (both per year and overall), monthly income, residence area urbanization level, and allergic diseases. Third, coding accuracy and financial incentives may lead to bias when ICD-9 CM codes are used for diagnosis in large insurance claims data for research. Although migraine is said to be the most common cause of primary headache in children, most children with headache were coded as “unspecified headache”. Some physicians were accustomed to give a diagnosis of headache rather than migraine for patients by only few visits. This implies that migraine may be underdiagnosed, particularly in young children and those with mild or infrequent symptoms. Children with mild symptoms, which may not be recognized by caregivers, may be only treated with over-the-counter medications. Doctors’ specializations may also interfere with the diagnosis. Therefore, we defined the studying children should have at least 3 times medical visits and were given diagnosis of “migraine” and/or “headache” in any diagnosis field during any inpatient or ambulatory claim process to capture more cases of migraine. Fourth, the individual exposure assessment was not conducted for each child. Air pollutants data collected from the government monitor station in the residential area were used instead. There are 74 ambient air quality-monitoring stations distributed all over the island of Taiwan. Most children attend the school near the residential area. Thus, the pollution levels to which they are exposed daily are approximately the average levels measured by the three stations near their resident area. In this study, we used the average levels measured for 12 years. Our data showed that risks associated with pollutant levels were consistent between boys and girls and between age groups. Fifth, weather change was previously mentioned as a trigger of migraine [37]. Because Taiwan has a subtropical climate, it does not have four distinct seasons; thus, climate should not be responsible for changes in prevalence throughout the year.

## 5. Conclusions

In conclusion, we herein demonstrate that long-term exposure to ambient air pollutants and relatively high ambient temperature are associated with recurrent headaches not only in adults but also in children. With this study, we hope to facilitate the implementation of an appropriate policy of public health for monitoring and further improving air quality—for the health of our children, who are the backbone of the future.

## Figures and Tables

**Figure 1 ijerph-17-09140-f001:**
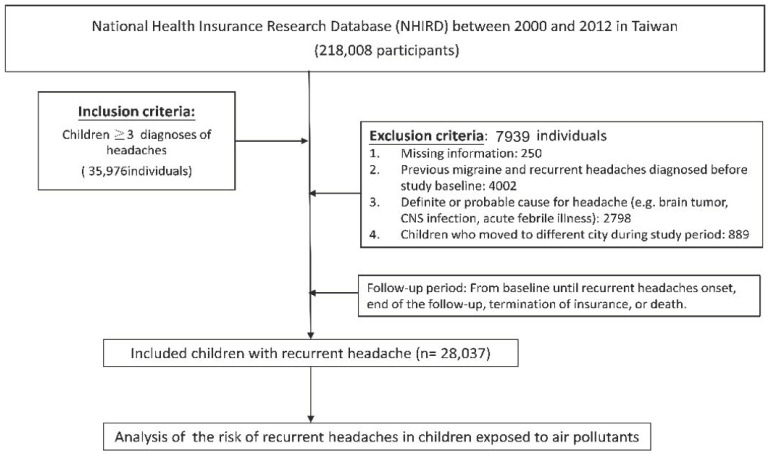
Flowchart of study design and study population selection.

**Figure 2 ijerph-17-09140-f002:**
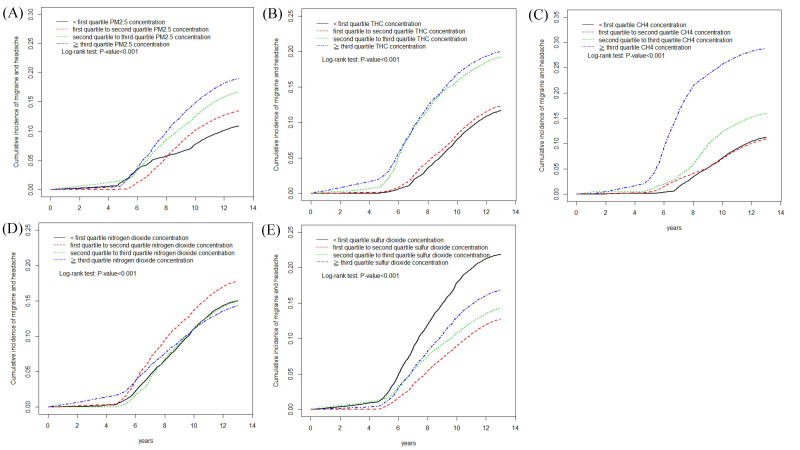
Kaplan–Meier curves of the accumulative incidence rate of recurrent headaches during the follow-up period among the four quartiles of each air pollutant. (**A**) PM_2.5_ (**B**) THC (**C**) CH_4_ (**D**) SO_2_ (**E**) NO_2_**.**

**Figure 3 ijerph-17-09140-f003:**
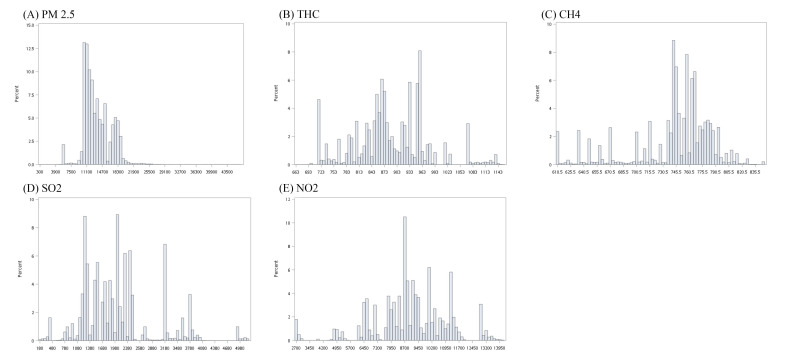
The distribution of annual air pollution exposures during the follow up time. (**A**) PM_2.5_ (**B**) THC (**C**) CH_4_ (**D**) SO_2_ (**E**) NO_2_.

**Table 1 ijerph-17-09140-t001:** Sociodemographic data of the study cohort.

*N* = 218,008		*n*	%
Gender	Boys	113,364	52.0
	Girls	104,644	48.0
Age, years	mean, SD	6.01	2.98
Monthly income (NTD) †	<15,000	187,199	85.9
	15,000−19,999	23,668	10.9
	≥20,000	7141	3.28
Urbanization level	1 (highest)	74,591	34.2
	2	69,003	31.7
	3	40,924	18.8
	4 (lowest)	33,490	15.4
Outcome			
recurrent headaches	Yes	28,037	12.9
Follow-up time, years	mean, SD	10.7	2.60

† Monthly income, new Taiwan Dollar (NTD), 1 NTD is equal to 0.03 USD. The urbanization level was categorized by the population density of the residential area into 4 levels, with level 1 as the most urbanized and level 4 as the least urbanized.

**Table 2 ijerph-17-09140-t002:** The risk of recurrent headaches in children exposed to air pollutants stratified by quartile of annual average concentration in Cox proportional hazard regression.

	IR	cHR	(95%CI)	aHR †	(95%CI)
Fine particulate matter (PM_2.5_)					
Quartile 1, <11,120 μg/m^3^	8.62	Reference group		Reference group	
Quartile 2, 11,120–12,652 μg/m^3^	11.0	1.30	(1.26, 1.35) **	1.29	(1.25, 1.34) **
Quartile 3, 12,652–15,056 μg/m^3^	13.5	1.59	(1.54, 1.65) **	1.57	(1.51, 1.62) **
Quartile 4, >15,056 μg/m^3^	15.0	1.78	(1.72, 1.84) **	1.75	(1.69, 1.81) **
Total hydrocarbons (THC)					
Quartile 1, <835 ppm	8.06	Reference group		Reference group	
Quartile 2, 835–877 ppm	8.40	1.05	(1.01, 1.09) *	1.08	(1.04, 1.12) **
Quartile 3, 877–949 ppm	16.9	2.26	(2.18, 2.34) **	2.41	(2.33, 2.49) **
Quartile 4, >949 ppm	16.3	2.25	(2.17, 2.33) **	2.50	(2.41, 2.59) **
Methane (CH_4_)					
Quartile 1, <735 ppm	8.93	Reference group		Reference group	
Quartile 2, 735–754 ppm	6.82	0.76	(0.73, 0.79) **	0.78	(0.75, 0.81) **
Quartile 3, 754–770 ppm	11.0	1.27	(1.22, 1.32) **	1.29	(1.24, 1.33) **
Quartile 4, >770 ppm	24.1	3.16	(3.06, 3.26) **	3.22	(3.11, 3.33) **
Sulfur dioxide (SO_2_)					
Quartile 1, <1346 ppb	13.7	Reference group		Reference group	
Quartile 2, 1346–1914 ppb	10.4	0.75	(0.72, 0.77) **	0.79	(0.76, 0.81) **
Quartile 3, 1914–2338 ppb	11.7	0.87	(0.84, 0.90) **	0.89	(0.86, 0.92) **
Quartile 4, >2338 ppb	12.3	0.90	(0.87, 0.92) **	0.93	(0.90, 0.96) **
Nitrogen dioxide (NO_2_)					
Quartile 1, <7896 ppb	11.8	Reference group		Reference group	
Quartile 2, 7896–8894 ppb	12.0	1.02	(0.99, 1.05)	1.07	(1.03, 1.10) **
Quartile 3, 8894–10,214 ppb	11.5	0.99	(0.96, 1.02)	1.05	(1.01, 1.09) *
Quartile 4, >10,214 ppb	12.9	1.13	(1.10, 1.17) **	1.23	(1.19, 1.27) **

IR, incidence rate (per 1000 person–years); cHR, crude hazard ratio; aHR, adjusted hazard ratio; CI, confidence interval. The annual average air pollutant concentrations were categorized into 4 groups based on quartiles for each air pollutant. † Adjusted HR, adjusted for age, sex, monthly income, urbanization level of residence, number of consultations/visits with a physician per year, and allergy diseases. * *p* < 0.01, ** *p* < 0.001.

**Table 3 ijerph-17-09140-t003:** The risk of recurrent headaches in children stratified by age, gender, and exposed to air pollutants stratified by quartile of annual average concentration in Cox proportional hazard regression.

Variables	Boys		Girls		Age ≤ 6		Age > 6	
	aHR †	(95%CI)	aHR †	(95%CI)	aHR †	(95%CI)	aHR †	(95%CI)
Fine particulate matter (PM_2.5_)								
Quartile 1, <11,120 μg/m^3^	Reference group		Reference group		Reference group		Reference group	
Quartile 2, 11,120–12,652 μg/m^3^	1.31	(1.24, 1.39) **	1.28	(1.22, 1.35) **	1.35	(1.29,1.42) **	1.21	(1.14, 1.29) **
Quartile 3, 12,652–15,056 μg/m^3^	1.59	(1.51, 1.68) **	1.55	(1.47, 1.62) **	1.61	(1.54, 1.68) **	1.50	(*1.41, 1.59) **
Quartile 4, >15,056 μg/m^3^	1.75	(1.67, 1.84) **	1.75	(1.67, 1.84) **	1.78	(1.70, 1.85) **	1.72	(1.62, 1.82) **
Total hydrocarbons (THC)								
Quartile 1, <835 ppm	Reference group		Reference group		Reference group		Reference group	
Quartile 2, 835–877 ppm	1.09	(1.03, 1.16) **	1.06	(1.01, 1.12) *	1.10	(1.05, 1.16) **	1.02	(0.95, 1.09)
Quartile 3, 877–949 ppm	2.60	(2.47, 2.74) **	2.24	(2.14, 2.35) **	2.38	(2.29, 2.48) **	2.48	(2.34, 2.63) **
Quartile 4, >949 ppm	2.77	(2.63, 2.93) **	2.27	2.16, 2.39) **	2.52	(2.41, 2.64) **	2.50	(2.35, 2.66) **
Methane (CH_4_)								
Quartile 1, <735 ppm	Reference group		Reference group		Reference group		Reference group	
Quartile 2, 735–754 ppm	0.78	(0.74, 0.83) **	0.77	(0.73, 0.81) **	0.76	(0.73, 0.80) **	0.81	(0.75, 0.87) **
Quartile 3, 754–770 ppm	1.35	(1.28, 1.42) **	1.23	(1.17, 1.30) **	1.31	(1.25, 1.37) **	1.25	(1.17, 1.34) **
Quartile 4, >770 ppm	3.50	(3.33, 3.67) **	2.98	(2.85, 3.12) **	3.15	(3.03, 3.28) **	3.37	(3.19, 3.57) **
Sulfur dioxide (SO_2_)								
Quartile 1, <1346 ppb	Reference group		Reference group		Reference group		Reference group	
Quartile 2, 1346–1914 ppb	0.80	(0.77, 0.85) **	0.77	(0.73, 0.81) **	0.77	(0.74, 0.81) **	0.81	(0.76, 0.86) **
Quartile 3, 1914–2338 ppb	0.91	(0.87, 0.96) **	0.87	(0.83, 0.91) **	0.87	(0.83, 0.91) **	0.93	(0.88, 0.99) **
Quartile 4, >2338 ppb	0.94	(0.90, 0.98) **	0.92	(0.88, 0.96) **	0.93	(0.89,0.97) *	0.92	(0.88, 0.97) **
Nitrogen dioxide (NO_2_)								
Quartile 1, <7896 ppb	Reference group		Reference group		Reference group		Reference group	
Quartile 2, 7896–8894 ppb	1.06	(1.01, 1.12) *	1.07	(1.02, 1.12) **	1.15	(1.10, 1.20) **	0.94	(0.88, 0.99) *
Quartile 3, 8894–10,214 ppb	1.11	(1.05, 1.17) **	1.00	(0.95, 1.05)	1.16	(1.11, 1.21) **	0.90	(0.85, 0.96) **
Quartile 4, >10,214 ppb	1.28	(1.22, 1.35) **	1.18	(1.12, 1.24) **	1.21	(1.15, 1.26) **	1.26	(1.20, 1.34) **

IR, incidence rate (per 1000 person–years); cHR, crude hazard ratio; aHR, adjusted hazard ratio; CI, confidence interval. The annual average air pollutant concentrations were categorized into 4 groups based on quartiles for each air pollutant. † Adjusted HR, adjusted for age, sex, monthly income, urbanization level of residence, number of consultations/visits with a physician per year, and allergy diseases. * *p* < 0.01, ** *p* < 0.001.

**Table 4 ijerph-17-09140-t004:** Association between ambient temperature with interaction of ambient air pollutants and risks for recurrent headaches by Cox proportional hazard regression analysis.

Ambient Air Pollutants	Ambient Temperature	IR	aHR † (95% CI)	*p*-Value for Interaction
Fine particulate matter (PM_2.5_)				<0.001
<Median	<Median	9.24	1.00 (Reference)	
<Median	≥Median	10.8	1.20 (1.15, 1.24) *	
≥Median	<Median	14.4	1.61 (1.53, 1.70) *	
≥Median	≥Median	14.3	1.55 (1.51, 1.60) *	
Total hydrocarbons (THC)				<0.001
<Median	<Median	6.97	1.00 (Reference)	
<Median	≥Median	8.93	1.26 (1.21, 1.31) *	
≥Median	<Median	13.4	2.11 (2.02, 2.21) *	
≥Median	≥Median	18.6	3.21 (3.08, 3.34) *	
Methane (CH_4_)				<0.001
<Median	<Median	7.5	1.00 (Reference)	
<Median	≥Median	7.97	1.04 (1.00, 1.08)	
≥Median	<Median	14.2	1.98 (1.90, 2.07) *	
≥Median	≥Median	18.6	2.74 (2.65, 2.84) *	
Sulfur dioxide (SO_2_)				<0.001
<Median	<Median	11	1.00 (Reference)	
<Median	≥Median	12.4	1.13 (1.09, 1.17) *	
≥Median	<Median	9.35	0.86 (0.82, 0.90) *	
≥Median	≥Median	14.4	1.36 (1.31, 1.42) *	
Nitrogen dioxide (NO_2_)				<0.001
<Median	<Median	9.58	1.00 (Reference)	
<Median	≥Median	13.0	1.38 (1.33, 1.44) *	
≥Median	<Median	10.3	1.16 (1.11, 1.21) *	
≥Median	≥Median	13.4	1.54 (1.48, 1.60) *	

IR, incidence rate (per 1000 person–years); aHR, adjusted hazard ratio; CI, confidence interval. The ambient air pollutants and ambient temperature were all categorized into 2 groups based on the median value of annual average. † Adjusted HR, adjusted for age, sex, monthly income, urbanization level of residence, number of consultations/visits with a physician per year, and allergy diseases. * *p* < 0.001.

**Table 5 ijerph-17-09140-t005:** Comparisons of differences in recurrent headaches incidences and associated HRs in participants exposed to various annual average concentrations of air pollutants.

Pollutant Levels	cHR	95%CI	aHR †	95%CI
Fine particulate matter (PM_2.5_)	1.000	(1.000, 1.000) *	1.000	(1.000, 1.001) *
Total hydrocarbons (THC)	1.003	(1.003, 1.003) *	1.004	(1.004, 1.004) *
Methane (CH_4_)	1.010	(1.010, 1.010) *	1.010	(1.010, 1.010) *
Sulfur dioxide (SO_2_)	1.000	(1.000, 1.000) *	1.000	(1.000, 1.001) *
Nitrogen dioxide (NO_2_)	1.000	(1.000, 1.000) *	1.000	(1.000, 1.001) *

IR, incidence rate (per 1000 person–years); cHR, crude hazard ratio; aHR, adjusted hazard ratio; CI, confidence interval. The annual average air pollutant concentrations were categorized into 4 groups based on quartiles for each air pollutant. † Adjusted HR, adjusted for age, sex, monthly income, urbanization level of residence, number of consultations/visits with a physician per year, and allergy diseases * *p* < 0.001.

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
