# Peer review of "Long-Term Ambient Air Pollutant Exposure and Risk of Recurrent Headache in Children: A 12-Year Cohort Study"

_ijerph, 2020, doi:10.3390/ijerph17239140_

Round 1
Reviewer 1 Report
This study examined the association between air pollution and childhood migraines using a retrospective cohort approach. Although the topic of this study is interesting and the sample size is large enough, there are still some serious problems need to be solved. The individual exposure assessment is very crucial in this study, however, authors only use air pollutants data collected from the three nearest stations for interpolation calculation, and do not give the evaluation criterion for interpolation effect. We do not consider this a rigorous scientific practice and doubt the accuracy of the interpolation results.
In this study, despite analyzing air pollution exposure evaluations as categorical variables, authors did not further explore air pollution exposure effects from various aspects, for example, its dose-effect relationship analysis, effects in different gender group, and so on. Thus, this study lacked some further analysis.
Additionally, there are mistakes found in the section of tables and figures. In discussion, there is no explanation for some negative results of this study, for example, SO2 exposure have no effect on migraine and recurrent headache.
Based on above points, I don’t think this manuscript should be considered to be acceptable.
Author Response
Manuscript ID: ijerph-992739 R1
Dear editor and reviewers
Thank you very much for your comments and suggestion on the manuscript,
entitled “Long-term ambient air pollutant exposure and risk of recurrent headache in children: A 12-year cohort study”. We are grateful to have the opportunity to revise this article. All the comments we received on this study have been taken into account improving the quality of the manuscript. The changes were highlighted in the revised
manuscript. We hope that these changes to the manuscript will facilitate the
decision to publish this study in your journal.
#Reviewer 1
- This study examined the association between air pollution and childhood migraines using a retrospective cohort approach. Although the topic of this study is interesting and the sample size is large enough, there are still some serious problems need to be solved. The individual exposure assessment is very crucial in this study, however, authors only use air pollutants data collected from the three nearest stations for interpolation calculation, and do not give the evaluation criterion for interpolation effect. We do not consider this a rigorous scientific practice and doubt the accuracy of the interpolation results.
Response: We appreciate your comments.
Ambient air monitoring of monthly average data for SO2, NO2, THC, CH4, and PM2.5 were collected from 74 ambient air quality-monitoring stations distributed all over Taiwan during 1998–2010. Concentrations of each pollutant are measured hourly—CO by nondispersive infrared absorption, NO2 by chemiluminescence, SO2 by ultraviolet fluorescence, THC and CH4 by flame ionization detector, and PM2.5 by beta-gauge—and are reported hourly. We identified the map coordinates of the monitoring stations and air pollution sources. Above descriptions were written in study method.
Lots of studies using above methods have been published in high quality journals, including your journal. (Ref: Environment international 94 (2016): 495-499; Environmental Pollution 254 (2019): 113031; Environmental Pollution 261 (2020): 114154; International Journal of Environmental Research and Public Health 15.12 (2018): 2860, etc.)
- In this study, despite analyzing air pollution exposure evaluations as categorical variables, authors did not further explore air pollution exposure effects from various aspects, for example, its dose-effect relationship analysis, effects in different gender group, and so on.
Response: We did dose-effect relationship analysis by continuous variables in original manuscript Table 4. (Please refer to revised manuscript Table 5.)
- …authors did not further explore air pollution exposure effects from various aspects, for example, effects in different gender group, and so on. Thus, this study lacked some further analysis.
Response: In our revised manuscript, we have added age-specific and gender-specific subgroup analysis in Table 3.
- Additionally, there are mistakes found in the section of tables and figures.
Response: We have rechecked all the tables and figures. We consider these mistakes occurred due to upload errors.
- In discussion, there is no explanation for some negative results of this study, for example, SO2 exposure have no effect on migraine and recurrent headache. Based on above points, I don’t think this manuscript should be considered to be acceptable.
Response: We have added some explanations why SO2 exposure have no effect on migraine and recurrent headache in the revisd manuscript [Discussion section, paragraph 8], 3 new references [38-40].
[Discussion section, paragraph 8]
“On the other hand, this study did not reveal positive association between ambient
concentration of SO2 and risks for recurrent headaches in children. Interestingly,
Szyszkowicz et al. conducted a study in Canada and found that SO2 exposure had
a positive correlation for migraine in female adults during warm seasons [38]. But another two studies in Asia did not find significant association between SO2 exposure and migraine development, whereas the other air pollutants were found to trigger migraine particularly on relatively hot and cool days [39,40]. Based on the above, we speculate because of the nature of sulfur dioxide-- a molecular compound that is heavier than air, its chemical properties are highly related to temperature and climate. Therefore, people who live in high latitude area with continental climate, year-round low temperature days, poorly ventilated and enclosed environment, are susceptible to the harmful influence of SO2 exposure. Taiwan is probably spared the damage because of its island type, subtropical climate. However, further studies are needed to confirm this presumption.”

Reviewer 2 Report
Introduction
- Line 36 - Migraines are not characterized by recurrent headaches. Please review ICHD3 beta classification and diagnosis of migraine. Also episodic tension-type and episodic migraines have different diagnostic criteria than chronic tension-type and chronic migraines. If the ICD-9-CM code is used for migraine/headache, please define this further that this study was conducted during time period when ICD9 diagnostic codes used and unclear if ICHD3 beta classification of headache disorders were utilized by physicians in Taiwan. Because ICHD3 criteria must be used for correct diagnosis, the headache field may not find this article as robust evidence. When identifying migraine/headaches as being more than 3 times diagnosed with the ICD9 code, please discuss this is the best method that you used (due to some of limitations you mentioned in discussion) to provide clarity to the headache expert reader.
Overall, the migraine/headache diagnosis is a large limitation but this was discussed thoroughly in the discussion. However, wanted to bring that to light in the forefront of the article for the specialized headache doctors that may not find the methods good enough to read further.
2. Great paragraph regarding minimal studies on pediatric triggers and migraines!
This study needed to be done and am so grateful for the authors to research this particular trigger for pediatric headaches.
Author Response
#Reviewer 2
- Line 36 - Migraines are not characterized by recurrent headaches. Please review ICHD3 beta classification and diagnosis of migraine. Also episodic tension-type and episodic migraines have different diagnostic criteria than chronic tension-type and chronic migraines. If the ICD-9-CM code is used for migraine/headache, please define this further that this study was conducted during time period when ICD9 diagnostic codes used and unclear if ICHD3 beta classification of headache disorders were utilized by physicians in Taiwan. Because ICHD3 criteria must be used for correct diagnosis, the headache field may not find this article as robust evidence. When identifying migraine/headaches as being more than 3 times diagnosed with the ICD9 code, please discuss this is the best method that you used (due to some of limitations you mentioned in discussion) to provide clarity to the headache expert reader. Overall, the migraine/headache diagnosis is a large limitation but this was discussed thoroughly in the discussion. However, wanted to bring that to light in the forefront of the article for the specialized headache doctors that may not find the methods good enough to read further.
Response: We appreciate your insightful comments.
Indeed, we admit that under the standard definition of migraine in ICHD3 criteria, many of our study subjects who were coding with "migraine" by ICD-9-CM may not be real migraine patients. However, given that core spirit of this study intended to convey the concept that "recurrent headaches" are likely to develop under the surrounding of "air pollution". Practically in clinical setting, we believe that if patients were coded as migraine (ICD-9-CM 346), it implied they might have a real migraine, or at least "recurrent headaches“ and those who had≧3 times diagnoses of ICD-9-CM code 346 and/or 784.0 (headaches) likewise. In consideration of your suggestion as well as to prevent misunderstanding, we decided to change our title from “Long-term ambient air pollutant exposure and risk of migraine and recurrent headache in children: A 12-year cohort study” to “Long-term ambient air pollutant exposure and risk of recurrent headache in children: A 12-year cohort study”. We also replaced all the term “migraine” by “recurrent headaches” across the text and only preserve the appropriate ones.
We have added more explanations in the "method section" "2.2. Study population, outcome of interest, endpoints, and confounding factors" to clarify our study method.
" National Health Insurance (NHI) data is a useful tool for massive epidemiological investigation but it has some systemic problems regarding accuracy when it targets to specific disease (especially disease diagnosed by clinical diagnostic criteria, e.g. migraine). The clinical diagnostic diseases obtained from ICD-9-CM from NHI data were generated by many different physicians individually and most of them were not specialist. Migraine-- characterized by recurrent headaches, were often confused used by physicians who were not specialists in Neurology and intend to describe "recurrent headaches" in their clinical setting. In view of this, we defined migraine/headaches in the present study as as≧3 times diagnoses of ICD-9-CM code 346 and/or 784.0 in any diagnosis field during any inpatient or ambulatory claim process during study period. Despite less than ideal, we believe this is the best method to select our target patients.
The flowchart of study design and study population selection was listed in Figure 1."
- Great paragraph regarding minimal studies on pediatric triggers and migraines! This study needed to be done and am so grateful for the authors to research this particular trigger for pediatric headaches.
Response: We appreciate your positive feedback. We are grateful to have the opportunity to revise this article. All the comments we received on this study have been taken into account improving the quality of the manuscript.
Reviewer 3 Report
Dear Authors and Editors
Hong et al. is a very though, interring study to investigate the association of levels of ambient air pollution with the incidence and the risk of migraine/headaches in Taiwan children.
Strength:
- Nationwide data of 12 years from Taiwan
- A literature review performed appropriately to identify missing links
- Thoroughly explained methods and results
- Balanced discussion
- The limitations are identified and explained well
My comments are below.
- Methods: Study population derivation with inclusion and exclusion criteria are not clear, you may add a figure showing that and STROBE protocol also requires that.
- Table 2 and 4 p values are missing
- Figure 2 should be described in footnote
Thank you
Author Response
Manuscript ID: ijerph-992739 R1
Dear editor and reviewers
Thank you very much for your comments and suggestion on the manuscript,
entitled “Long-term ambient air pollutant exposure and risk of recurrent headache in children: A 12-year cohort study”. We are grateful to have the opportunity to revise this article. All the comments we received on this study have been taken into account improving the quality of the manuscript. The changes were highlighted in the revised
manuscript. We hope that these changes to the manuscript will facilitate the
decision to publish this study in your journal.
#Reviewer 3
- Methods: Study population derivation with inclusion and exclusion criteria are not clear, you may add a figure showing that and STROBE protocol also requires that.
Response: We have made a flowchart of study design and study population selection in figure 1.
- Table 2 and 4 p values are missing
Response: We have corrected the errors in revised manuscript accordingly.
- Figure 2 should be described in footnote
Response: We have corrected the errors in revised manuscript accordingly.
Round 2
Reviewer 1 Report
Although the author has made some modifications to the content of the article as suggested, it is obvious that the main question we raised, namely the assessment of individual pollution exposure, has not been answered well . I think the method of using the data from three stations to predict the pollutant level at a certain geographic coordinates is not scientific enough,and the paper also lacks the accuracy evaluation index of interpolation method. So, we don’t think this manuscript should be considered to be acceptable.
Author Response
Response to Reviewers’ comments
Manuscript ID: ijerph-992739 R2
Title: Long-term ambient air pollutant exposure and risk of recurrent headaches in children: A 12-year cohort study
Dear Editor and Reviewers:
Thank you very much for your comments and suggestion for our study.
We are grateful to have the opportunity to revise this article. All the comments we received on this study have been taken into account to improve our study. We have completed the revision based on Reviewers’ comments point by point. The changes were highlighted in the revised manuscript. We hope that these changes to the manuscript will facilitate the decision to publish this study in your journal.
#Reviewer 1 (Round 1)
- This study examined the association between air pollution and childhood migraines using a retrospective cohort approach. Although the topic of this study is interesting and the sample size is large enough, there are still some serious problems need to be solved. The individual exposure assessment is very crucial in this study, however, authors only use air pollutants data collected from the three nearest stations for interpolation calculation, and do not give the evaluation criterion for interpolation effect. We do not consider this a rigorous scientific practice and doubt the accuracy of the interpolation results.
#Reviewer 1 (Round 2)
- Although the author has made some modifications to the content of the article as suggested, it is obvious that the main question we raised, namely the assessment of individual pollution exposure, has not been answered well. I think the method of using the data from three stations to predict the pollutant level at a certain geographic coordinates is not scientific enough,and the paper also lacks the accuracy evaluation index of interpolation method. So, we don’t think this manuscript should be considered to be acceptable.
Response: Thank you very much for the inspirational comment, which reminded us the issue for discussion in the study limitation.
We agree with the comment that the air pollution children exposed to was best measured individually using personal device for a period of time. Air pollution level changed from time to time. It takes a great amount of efforts to conduct the personal exposure measures for a large number of children, a very expensive study we are not capable to do. We, therefore, used the government monitored data of pollution to for interpolation calculation, simulating other population-based studies.
Taiwan is an island with 74 ambient air quality-monitoring stations distributed all over Taiwan. The measured hourly concentrations of pollutants at each station represent the levels residents exposed to in the area with the monitor station. For an example, Taipei city is a metro-city with an area of 272 Km2 (105 mile2). There are 6 monitor stations in Taipei city, one station per 45 Km2 (or per 18 mile2) in average. Most children attend the school near their residential areas. Thus, by using the average level of the 3 nearest stations, the pollution levels they exposed to daily are approximately the levels measured by the stations near their resident. In this study, we used the average levels measured for 12 years. Our data showed that risks associated with pollution levels are consistent between boys and girls, and between age groups. We believe that a study hypothesis is likely worth to be confirmed in other populations.
In our revision, we included the inspirational comment as one of our study limitations: “Fourth, the individual exposure assessment was not conducted for each child. Air pollutants data collected from the government monitor station in the residential area were used instead. There are 74 ambient air quality-monitoring stations distributed all over the island of Taiwan. Most children attending the school near the residential area. Thus, the pollution levels they exposed to daily are approximately the average levels measured by the 3 stations near their residents. In this study, we used the average levels measured for 12 years. Our data showed that risks associated with pollutant levels were consistent between boys and girls and between age groups.” (Please see lines 406-413.)
#Reviewer 1 (Round 1)
- In this study, despite analyzing air pollution exposure evaluations as categorical variables, authors did not further explore air pollution exposure effects from various aspects, for example, its dose-effect relationship analysis, effects in different gender group, and so on.
Response: Thank you for the comment.
We have further conducted data analysis to compare risks between boys and girls in Table 3. Data analyses for Tables 2, 3 and 4 were based on categorical dose-levels of pollutants by quartiles. Most pollutants did demonstrate dose-response relationships. Table 3 shows findings by sex and age; results showed that most relationships were consistent between boys and girls and between younger and older children. In response to the comment, we further calculated the correlation coefficients between incidence rates and pollutant levels. Results of relationships with pollutants are: r=0.98 with PM2.5, r=0.68 with THC, r=0.52 with CH4, r=0.07 with SO2 and r=0.04 with NO2.
Dose-response effects are significant for PM2.5, THC and CH4. (Please see lines 255-258.)
